# Gold-Decorated Platinum and Palladium Nanoparticles as Modern Nanocomplexes to Improve the Effectiveness of Simulated Anticancer Proton Therapy

**DOI:** 10.3390/pharmaceutics13101726

**Published:** 2021-10-18

**Authors:** Bartosz Klebowski, Malgorzata Stec, Joanna Depciuch, Adrianna Gałuszka, Anna Pajor-Swierzy, Jarek Baran, Magdalena Parlinska-Wojtan

**Affiliations:** 1Institute of Nuclear Physics Polish Academy of Sciences, 31-342 Krakow, Poland; joanna.depciuch@ifj.edu.pl (J.D.); magdalena.parlinska@ifj.edu.pl (M.P.-W.); 2Department of Clinical Immunology, Jagiellonian University Medical College, 30-663 Krakow, Poland; malgorzata.stec@uj.edu.pl (M.S.); adrianna.galuszka@doctoral.uj.edu.pl (A.G.); mibaran@cyf-kr.edu.pl (J.B.); 3Jerzy Haber Institute of Catalysis and Surface Chemistry Polish Academy of Sciences, 20-239 Krakow, Poland; anna.pajor-swierzy@ikifp.edu.pl

**Keywords:** bimetallic nanoparticles, radiosensitizers, proton anticancer therapy, colon cancer cells, green chemistry, transmission electron microscopy, MTS test, flow cytometry

## Abstract

Noble metal nanoparticles, such as gold (Au NPs), platinum (Pt NPs), or palladium (Pd NPs), due to their highly developed surface, stability, and radiosensitizing properties, can be applied to support proton therapy (PT) of cancer. In this paper, we investigated the potential of bimetallic, c.a. 30 nm PtAu and PdAu nanocomplexes, synthesized by the green chemistry method and not used previously as radiosensitizers, to enhance the effect of colorectal cancer PT in vitro. The obtained nanomaterials were characterized by scanning transmission electron microscopy (STEM), selected area electron diffraction (SAED), energy-dispersive X-ray spectroscopy (EDS), UV-Vis spectroscopy, and zeta potential measurements. The effect of PtAu and PdAu NPs in PT was investigated on colon cancer cell lines (SW480, SW620, and HCT116), as well as normal colon epithelium cell line (FHC). These cells were cultured with both types of NPs and then irradiated by proton beam with a total dose of 15 Gy. The results of the MTS (3-(4,5-dimethylthiazol-2-yl)-5-(3-carboxymethoxyphenyl)-2-(4-sulfophenyl)-2H-tetrazolium) test showed that the NPs-assisted PT resulted in a better anticancer effect than PT used alone; however, there was no significant difference in the radiosensitizing properties between tested nanocomplexes. The MTS results were further verified by defining the cell death as apoptosis (Annexin V binding assay). Furthermore, the data showed that such a treatment was more selective for cancer cells, as normal cell viability was only slightly affected.

## 1. Introduction

Globally, colorectal cancer is a major health problem with an increasing incidence [1]. Cancer cells are often resistant to classical treatment, thus it is highly desirable to find new tools to fight this disease [2]. For colorectal cancer, the photon radiotherapy (with X- or gamma rays) was introduced to reduce the tumor volume before surgery or in palliative treatments [3,4]. However, a better irradiation effect can be achieved by proton radiotherapy (PT) with an accelerated proton beam. Compared to photons, which deposit their radiation doses close to the entrance into the body, the protons deposit much more energy at the end of the path [5].

The effect of PT can be further increased by using nanosized radiosensitizers. Nanotechnology offers sub-100 nm nanomaterials for many biomedical applications, such as: drug delivery systems, MRI contrast agents, treatment of fungal or bacterial infections, or as radiosensitizers in radiation-based anticancer therapies, including PT [6,7,8,9,10,11]. Noble metal nanoparticles, based on gold (Au NPs), platinum (Pt NPs), and palladium (Pd NPs), are characterized by highly developed surface, stability, good thermal properties, and the ability to generate reactive oxygen species (ROS) under the influence of applied radiation, which increases their biological effectiveness [12,13,14,15]. So far, the literature contains several papers on using in vitro Au, Pt, bismuth (Bi), gadolinium (Gd) NPs, as well as bimetallic nanocomplexes as potential radiosensitizers in PT [16,17,18,19,20]. The experimental studies correspond to a number of theoretical works using the Monte Carlo method to assess the improvement of PT by NPs [18,21,22,23]. To emphasize the potential of such metallic radiosensitizers, it is worth noting that platinum-based compounds (e.g., cisplatin, carboplatin, oxaliplatin, nedaplatin, and lobaplatin) have been successfully used for many years as chemotherapeutic agents, improving the effects of subsequent radiotherapy [24]. There is also interest in the application, alternatively, of gold or palladium-based compounds for this purpose [25,26]. 

Therefore, in this paper, the combined effect of proton irradiation on the selected colon cancer cells with different malignancy potential (SW480, SW620, HCT116), as well as normal colon epithelium cells (FHC) in the presence of the bimetallic PtAu and PdAu nanocomplexes was investigated. From the economic point of view, it was also interesting to show whether NPs with a palladium core (the synthesis of which is cheaper) will be as effective as NPs with a platinum core. Furthermore, until now, in the literature, no information is available on the application of Pd NPs to sensitize cells exposed to the proton beam.

The NPs analyzed in the current study were synthesized by green chemistry with the use of gallic acid. The characterization of nanocomplexes was done using scanning transmission electron microscopy (STEM). The crystalline nanostructure of PtAu and PdAu NPs were confirmed by selected area electron diffraction (SAED). Additionally, the chemical composition of bimetallic nanocomplexes was evaluated by energy-dispersive X-ray spectroscopy (EDS). The MTS ((3-(4,5-dimethylthiazol-2-yl)-5-(3-carboxymethoxyphenyl)-2-(4-sulfophenyl)-2H-tetrazolium) viability test and flow cytometry were used to assess the cell damage caused by PtAu and PdAu NPs, as well as NPs-assisted proton irradiation. The study design is schematically presented in Figure 1. 

## 2. Materials and Methods

### 2.1. Reagents and Chemicals

Hydrogen tetrachloroaurate trihydrate (HAuCl_4_∙3H_2_O), hydrogen hexachloro- platinate hexahydrate (H_2_PtCl_6_∙6H_2_O), palladium (II) chloride (PdCl_2_), and gallic acid were obtained from Sigma-Aldrich (Saint Louis, MO, USA). Chemical reagents were used without additional purification or modification.

### 2.2. Synthesis of Bimetallic Nanocomplexes

#### 2.2.1. Synthesis of Gold-Decorated Platinum Nanoparticles (PtAu NPs)

PtAu NPs were synthesized by the green chemistry method with gallic acid acting simultaneously as stabilizer and reducing agent. For this purpose, 250 µL of 0.01 M H_2_PtCl_6_ aqueous solution was added to 17.5 mL distilled water in a round-bottom flask. The reaction mixture was heated until boiling on a magnetic stirrer (350 rpm) and then 2 mL of 0.005 M freshly prepared gallic acid solution was placed into flask. The reaction was carried out for 40 min, observing slow color change of the solution from colorless to gray. Then, 150 µL of 0.01 M HAuCl_4_ solution was added to the mixture and the reaction was continued for 40 min. The color change to violet was observed just after the addition of the gold precursor.

#### 2.2.2. Synthesis of Gold-Decorated Palladium Nanoparticles (PdAu NPs)

In order to obtain H_2_PdCl_4_, it was necessary to treat PdCl_2_ with an equivalent of hydrochloric acid (HCl). The reaction was carried out analogously to that described in Section 2.2.1 with the difference that the H_2_PdCl_4_ was used instead of the H_2_PtCl_6_. After addition of gallic acid, the color of the solution slowly changed from colorless to gray-brown. In turn, when the HAuCl_4_ solution was poured into flask, the solution immediately turned violet-brown.

### 2.3. TEM Characterization

The morphology of the PtAu and PdAu NPs was determined by scanning transmission electron microscopy (STEM) combined with the high-angle annular dark-field detector (HAADF) working in conventional and high-resolution. SAED patterns were taken in the TEM mode. All these measurements were performed on an aberration-corrected FEI Titan electron microscope (Hillsboro, OR, USA) operating at 300 kV equipped with a FEG (field emission gun) cathode. EDS mappings were done using a FEI Talos TEM operating at 200 kV equipped with a FEG cathode and four in-column EDS detectors (Super EDS system) The particle size distribution was evaluated based on the STEM images taken from different areas of the TEM grids. The diameter of about 100 PtAu and PdAu NPs was measured as a distance between the two most distant points of these NPs.

### 2.4. UV-Vis Spectroscopy

The UV-Vis measurements were performed with a Multiskan SkyHigh spectrometer from Thermo Scientific (Waltham, MA, USA). The resolution was chosen to be 1 nm and the spectral range was from 300 nm to 900 nm.

### 2.5. Zeta Potential Measurements 

The zeta potential distribution of PtAu and PdAu NPs was determined by the microelectrophoretic method using Zetasizer Nano Series from Malvern Instruments (Worcestershire, UK). The Smoluchowski model was used in zeta potential measurements. Each value was obtained as an average of three subsequent runs of the instrument with at least 20 measurements. All experiments were performed in water at room temperature.

### 2.6. Cell Culture 

Colon cancer cell lines (SW480 and SW620—derived from the same patient, from primary tumor and metastatic lesion to lymph node, respectively) were obtained thanks to the courtesy of Professor Caroline Dive, Paterson Institute for Cancer Research, University of Manchester. Human colon carcinoma cell line HCT116 (from primary tumor) and normal colon epithelium cell line FHC (CRL-1831) were obtained from the American Type Culture Collection (ATCC, Manassas, VA, USA) and maintained according to the ATCC instructions. HCT116 cells were cultured in McCoy’s 5A medium (Gibco, Paisley, UK). SW480 and SW620 cells were cultured in DMEM (Dulbecco’s modified eagle medium) with high glucose (Corning, NY, USA). FHC cells were cultured in DMEM/F12 medium (Gibco) supplemented with 25 mM HEPES (4-2(hydroxyethyl)-1-piperazineethanesulfonic acid), 10 ng/mL cholera toxin, 0.005 mg/mL insulin, 0.005 mg/mL transferrin, 100 ng/mL hydrocortisone. All media were supplemented with 10% fetal bovine serum (FBS) and ciprofloxacin (10 µg/mL). The cells were cultured by bi-weekly passages in a 37 °C humidified atmosphere with 5% CO_2_ and regularly tested for *Mycoplasma* sp. contamination by PCR-ELISA kit (Roche, Mannheim, Germany) according to the manufacturer’s recommendations.

### 2.7. Proton Irradiation and Dosimetry 

Irradiations were performed in the Cyclotron Centre Bronowice, Institute of Nuclear Physics Krakow PAS, Poland. The proton therapy system installed in the Centre consists of an IBA Proteus C-235 cyclotron (IBA PT, Louvain-la-Neuve, Belgium) and two gantries equipped with scanning nozzles. In the pencil beam scanning (PBS) techniques, a narrow proton beam is deflected in two perpendicular directions, delivering the dose point by point to the whole target volume. Irradiations were performed using a monoenergetic field with an energy of 225 MeV and dimensions of 20 cm × 20 cm. The cells were irradiated at 1.1 cm water equivalent depth with a dose of 15 Gy. The gantry was set at 180°, which means that the beam direction was from the bottom to the top. The preparation of the experiment included dosimetry measurements performed with a Markus type ionization chamber calibrated in terms of dose absorbed to water. A radiation dose of 15 Gy was selected for our research, because this dose in non-toxic to our cell lines. After proton irradiation, each cell line was incubated for 18 h and then a survival MTS/annexin V-binding assay was performed.

### 2.8. MTS Cytotoxicity Assay 

The cytotoxic activity of PtAu and PdAu NPs against normal and cancer cells was determined using the 3-(4,5-dimethylthiazol-2-yl)-5-(3-carboxymethoxyphenyl)-2-(4-sulfo-phenyl)-2H-tetrazolium (MTS) assay (CellTiter 96^®^ Aqueous One Solution Cell Proliferation Assay, Promega, Madison, WI, USA). Briefly, the cells were cultured in flat-bottom 96-well plates (Sarstedt, Numbrecht, Germany) at a density 104/well in DMEM medium containing 10% FBS. After 48 h, 20 μL of PtAu and PdAu NPs solutions with different concentrations were added to the 100 μL medium with cells (final concentration of NPs 5–150 µg/mL). After an incubation period of 18 h, 20 µL of MTS dye solution were added per well. The quantity of formazan product, directly proportional to the number of living cells in culture, was detected by absorbance measurements at 490 nm with a 96-well plate reader (Spark^®^ Tecan, Mannedorf, Switzerland). Analogous measurements were also performed for incubation times of 3, 24, and 42 h. For further PtAu/PdAu NPs-assisted proton irradiation studies, a maximal concentration, not causing a decrease in cell viability by more than 15%, comparing to untreated groups, after 18 h incubation, was selected (75 µg/mL for SW480 and 100 µg/mL for SW620 and HCT116 cells). As for the FHC cell line (control), these cells were treated with both (75 and 100 µg/mL) PtAu and PdAu NPs concentrations allowing a reliable comparison of the effect of proton irradiation on normal and colon cancer cells. Each sample test was repeated three times. The description of the samples analyzed in this experiment is shown in Table 1.

### 2.9. Analysis of Cell Apoptosis 

Apoptosis of normal and cancer cells treated with PtAu/PdAu NPs and proton beam irradiation was evaluated by Annexin V binding assay (Fluorescein isothiocyanate (FITC) Annexin V apoptosis detection Kit I (BD Pharmingen, San Diego, CA, USA)) and flow cytometry analysis, as described previously [27]. Cells were plated in a 24-well plate at the density of 10^6^/well. After 48 h of incubation, the cells were exposed to PtAu/Pd Au NPs with the optimal concentrations (analogous to Section 2.8) and irradiated with protons. Then, the medium from each well was transferred in a pre-labeled separate centrifuge tube. Adherent cells from the same well were then trypsinized and transferred to the same tubes. All samples were centrifuged, and the supernatant discarded. Pellets were re-suspended in a solution containing Annexin V-FITC conjugated and incubated for 10 min in the dark prior to flow cytometry analysis. All analyses were performed on a FACS Calibur flow cytometer (BD Biosciences, Immunocytometry Systems, San Jose, CA, USA). The number of the acquired events was no less than 10,000. The results were analyzed using a FACS Diva v.8.1 software (BD Biosciences).

### 2.10. Statistical Analysis of Cell Viability Data

The obtained zeta potential values and MTS assay results are represented as the means ± SEM (standard error of the mean). The experimental data (for MTS test) were analyzed by one-way analysis of variance (ANOVA) followed by post-hoc Tukey test. A *p* value < 0.05 was considered as statistically significant. The data were analyzed and presented graphically using GraphPad Prism 8 Software.

## 3. Results and Discussion

### 3.1. Mechanism of PtAu and PdAu NPs Synthesis

To obtain the bimetallic NPs, the green chemistry method with gallic acid was applied. Gallic acid (3,4,5-trihydroxybenzoic acid) is a natural component of, e.g., green teas, grapes, or tomatoes. In our synthesis, it served as a reducing agent, as well as a stabilizer, preventing the agglomeration of NPs [28]. The undoubted advantage of gallic acid is not only its eco-friendliness, but also its low cost and a wide spectrum of activity, as a potential reducing agent of various noble metal precursors [20,29,30,31,32].

The proposed reaction mechanism of the PtAu and PdAu NPs synthesis is depicted in Figure 2. 

In the first step, the platinum or palladium precursor are reduced with gallic acid at the boiling point of water. This process is not fast, as can be seen from the slow color change of the reaction mixture. In turn, in the next step after the addition of chloroauric acid to the reaction mixture containing pre-synthesized Pt or Pd NPs, the rest of the reaction occurs rapidly, which results in an immediate color change of the solution. As demonstrated in previous work, this is due to the fact that there is a significant difference in the redox potentials of the platinum/palladium and gold precursors [20,31]. These differences in redox potential are the driving force behind this reaction and therefore, they take place quickly. It is also worth noting that the simultaneous placing of both precursors in the reaction vessel at the same time, followed by their reduction with gallic acid, would result in obtaining bimetallic nanoparticles consisting of a gold core and a platinum/palladium shell. The gold precursor, due to the much higher redox potential, would be reduced faster than the other, as demonstrated in our previous study [20].

### 3.2. TEM Characterization

The morphology, crystalline nanostructure, and chemical composition of PtAu and PdAu NPs were studied by electron microscopy. In Figure 3, STEM HAADF images of both types of nanocomplexes (Figure 3(a1,b1)) with the corresponding EDS maps (Figure 3(a2,b2)) of Au and Pd/Pt distribution are shown. We were able to distinguish individual NPs due to the Z-contrast differences of these NPs. From the EDS maps, it is clear that the Pt/Pd NPs are decorated with Au NPs; however, the shape of both of these complexes is irregular. Moreover, the Au NPs on the surface of PtAu/PdAu NPs are characterized not only by their size (5–20 nm), but also by their shape (we can observe, e.g., spherical and triangular-like nanogold). The shape and size of the Pt/Pd core also varies. Generally, the TEM photos show PtAu/PdAu NPs with a total diameter of approximately 30 nm. Due to the irregular shape, this diameter was conventionally measured as the distance between two most distant points of the NPs. From the EDS maps, it can also be seen that the Au does not cover the platinum/palladium core continuously. From the SAED pattern (Figure 3(a3,b3)), it was found that both types of nanocomplexes have a crystalline nanostructure, because sharp diffraction rings with bright spots were observed. The diffraction rings can be attributed to the (111), (200), (220), (331) and—for PdAu NPs—(222), and (400) lattice planes of Au, Pt, and Pd nanocrystals crystallized in the face-centered cubic structure [33,34]. Figure 3(a4,b4) shows the size distribution of the obtained PtAu and PdAu NPs. For PtAu NPs, a smaller size dispersion was observed compared to PdAu NPs. For PtAu NPs, the histogram resembles a Gaussian distribution, which was not found for the PdAu NPs. 

### 3.3. UV-Vis Spectroscopy

The optical properties of the synthesized nanocomplexes were investigated by UV-Vis spectroscopy, as shown in Figure 4.

In the UV-Vis spectra of both types of nanocomplexes, one peak at 740 nm was visible. The STEM images presented in Figure 3(a1,b1) show that Au NPs, which decorated Pt or Pd NPs, have spherical and triangular-like shapes. In the case of spherical Au NPs, the UV-Vis peak position should be around 540 nm, while triangular Au NPs show the absorption of light at a wavelength between 800 nm and 1300 nm [35]. Because there is a contribution of a mixture of spherical and triangular Au NPs (Figure 3(a1)), therefore the peak is located between the values corresponding to pure Au spheres and pure Au triangles. 

### 3.4. Zeta Potential Measurements

The zeta potential values of PtAu and PdAu NPs dispersions as a function of their pH are depicted in Figure 5. The zeta potential values for both types of nanoparticles are negative in the tested pH range. For PtAu NPs, the zeta potential values range from about −20 to −30 mV and do not seem to depend on the pH. In turn, for PdAu NPs these values range from −30 to −40 mV. However, for lower pH, a statistically significant increase in zeta potential value is visible. Relatively slight differences in the zeta potential of both types of NPs result from the use of a similar procedure for their synthesis. The negative values of the zeta potential of the obtained nanoparticles correspond to the measurements of other research groups, where the zeta potentials of metallic nanoparticles, stabilized with gallic acid, were also investigated [32,36].

### 3.5. Enhancement of Proton Irradiation Effect on Colon Cancer Cells by PtAu and PdAu NPs

#### 3.5.1. MTS Test

The cytotoxicity of bimetallic PtAu and PdAu NPs (Figure 6a–d) and their effect on the proton treatment of colon cancer cells (SW480, SW620, and HCT116) or normal colon epithelium cells (FHC) was evaluated using the MTS viability test (Figure 7a–c). The cytotoxicity of PtAu and PdAu NPs in different concentrations was measured after 18 h of cell incubation with nanocomplexes. The SW480 cancer cells turned out to be most sensitive for the action of the studied NPs. The high sensitivity of this cell line to various factors, comparing to other cancer cell lines, e.g., SW620, was already demonstrated [37,38]. Both types of NPs did not differ significantly in their activity against the tested cell lines. Based on these results, the maximum concentrations of NPs (conventionally called the optimal concentration), not causing a significant decrease in cell viability (i.e., more than 15%, comparing to untreated groups), was selected for further studies. The optimal concentration of NPs for the SW480 cells was considered to be 75 µg/mL, for the other two cancer cell lines (SW620 and HCT116)—100 µg/mL. In the case of normal FHC cells, they were cultured with both PtAu/PdAu NPs concentrations to reliably compare the proton irradiation effect on the cancer and normal cells.

The effect of PT on cell viability after pretreatment with PtAu and PdAu nanocomplexes (incubation time = 18 h) was then determined (Figure 7). The total radiation dose of 15 Gy was chosen as not causing cell damage by itself. The results showed that PtAu and PdAu NPs-assisted PT was more effective than classic PT. The addition of NPs in low concentrations significantly enhanced the effect of the treatment. The viability of C@PtAu NPs@PR^+^ was 52%, 52%, and 64% of control for SW480, SW620, and HCT116 cells, respectively. In turn, the survival of C@PdAu NPs@PR^+^ was estimated to be 58%, 54%, and 70% for SW480, SW620, and HCT116 cell lines, respectively. Among the cancer cell lines, HCT116 cells showed the poorest response to such combination treatment, corroborating earlier reports [39,40,41]. Moreover, no statistically significant difference was demonstrated in respect to the effect of PtAu and PdAu NPs on cancer cells. This indicates that both types of NPs could be used as potential radiosensitizers; however, taking into account that the production costs of Pd NPs are clearly lower than for Pt NPs, the Pd NPs seem to be advantageous for an economical reason. For other incubation times (3, 24, and 42 h), the results of the MTS test are presented in the Appendix A.

#### 3.5.2. Flow Cytometry

Apoptosis of cells exposed to the combined treatment was quantified by Annexin V staining and flow cytometry analysis. At the early phase of apoptosis, phosphatidylserine (PS) is externalized and the use of fluorescently-labeled Annexin V, which binds specifically to PS, allows for the detection of dying cells. The results of the flow cytometry analysis of apoptotic cells after the specific treatment are summarized in Figure 8a–d. Dot-plots show live (Annexin V-negative) and apoptotic (Annexin V-positive) cells. In general, these results correlate with those obtained by the MTS test, although a slightly higher amount of dead cells was observed in the case of the SW480 cell line. In this context, it is worth noting that for this cell line, the cell viability in the control group (without any treatment) was lower, comparing to the other cell lines. For the cancer cells, a significant increase in the level of apoptotic cells was observed as a result of combined PT (36–82%, depending on cell lines—the weakest effect for HCT116). Only a small population of apoptotic cells was detected in the case of FHC cells (c.a., 20% in any case).

Such a selectivity of PtAu/PdAu-assisted PT may result from the fact that gallic acid was used for the NPs synthesis process, as this compound is known to possess anticancer activity [42,43,44]. In the case of bimetallic nanoparticles, their high radiosensitizing activity may result from the combined biological effects of Au and Pt/Pd. As noted in Section 3.2., the core of bimetallic Pt/Pd NPs is not continuously covered with gold. Accordingly, cells are in contact with both metals and their mechanism of radiosensitization may be different. This concept would explain the radiosensitizing activity of the NPs, although based on the Bethe–Bloch formula, the NPs enhancement of proton beam is nearly independent of the Z atomic number [45]. It seems that different mechanisms may be responsible for enhancing the radiosensitizing effect of NPs. Noble metal NPs under the influence of high-energy irradiation generate ROS, which result in a double-strand breakage of DNA and cell apoptosis [46]. It was shown that Au NPs, Pt NPs, and bismuth (Bi) NPs treated with high-energy protons generate ROS and cause apoptosis of HCT116 cells [15]. In turn, Kim et al. showed that, as a result of combined therapy (Au NPs + X-rays), the forming ROS enhance the colon cancer cell (CT26, mouse model) death [47]. A similar effect was observed for octaarginine-modified Au NPs [48]. ROS can also affect other biomolecules, e.g., lipids, causing their peroxidation. Combined therapy with nanoparticles (mainly Au NPs) can also disrupt the cell cycle, resulting in cell death [15]. A biological mechanism, which can also have a negative effect on cells is the so-called, bystander effect”. This effect is observed in cells not exposed to direct irradiation, which however can receive signals from neighboring irradiated cells and ultimately the proliferation of such cells can be inhibited or the cells die by apoptosis. There are also reports in the literature emphasizing the importance of bismuth, silver, and Au NPs in enhancing the bystander effect” [49,50,51]. Some reports indicate the dependence of the zeta potential value on the cellular uptake of NPs [52,53,54]. Generally, it is accepted that NPs with higher zeta potential are characterized by increased cellular uptake, and they would potentially be better radiosensitizers. Moreover, positively charged NPs, due to the more favorable electrostatic interactions with negatively charged cell membrane, are better uptaken by cells than neutral or negatively charged NPs [52,53]. However, for both types of NPs used, the zeta potentials differed only slightly, and therefore, presumably had no effect on the cellular uptake. 

## 4. Conclusions

In summary, we have obtained bimetallic, 30 nm Pt NPs and Pd NPs decorated with Au NPs. For this purpose, the green chemistry synthesis method was used. The use of green chemistry has positive effect not only on the environment, but also on the biocompatibility of the obtained nanomaterials. The application of PtAu and PdAu NPs as potential radiosensitizers in PT of colorectal cancer has shown that such combined approach results in a significant inhibition of cancer cell proliferation and viability. On the other hand, normal cells were affected to a much lesser extent, which is presumably due to the selectively acting gallic acid used in the NPs synthesis process. We hope that in the near future, such nanomaterials will become a powerful clinical tool, supporting PT and other types of radiation-based anticancer therapies.

## Figures and Tables

**Figure 1 pharmaceutics-13-01726-f001:**
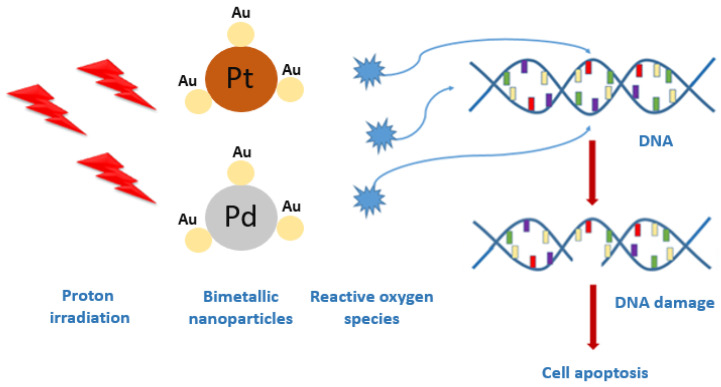
General scheme of the investigation procedure.

**Figure 2 pharmaceutics-13-01726-f002:**
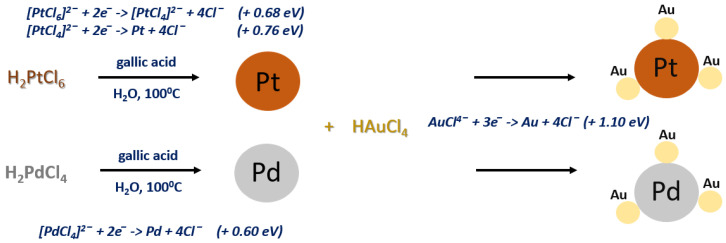
Proposed scheme of the PtAu and PdAu NPs synthesis.

**Figure 3 pharmaceutics-13-01726-f003:**
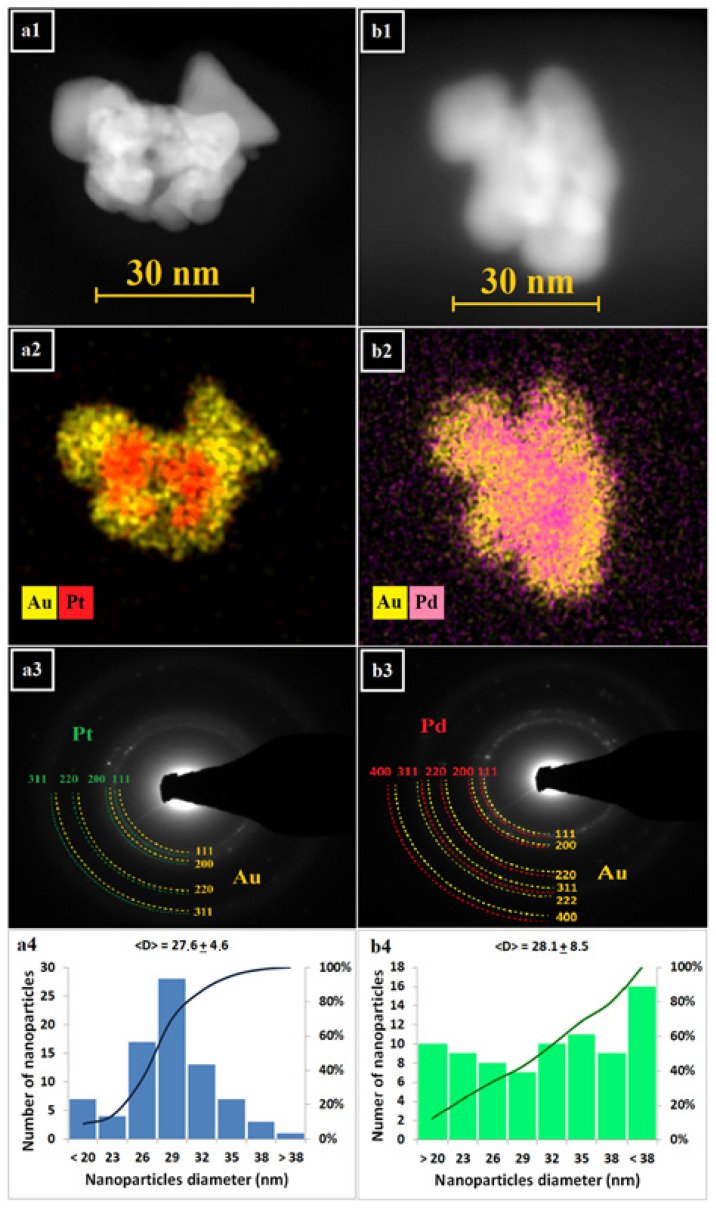
Scanning transmission electron microscopy (STEM), high-angle annular dark-field (HAADF) detector images (**a1**,**b1**), corresponding energy dispersive X-ray spectroscopy (EDS) distribution maps (**a2**,**b2**), selected area electron diffraction (SAED) patterns (**a3**,**b3**) and size distribution (histograms and distribution curves; (**a4**,**b4**) of PtAu (**a**) and PdAu NPs (**b**).

**Figure 4 pharmaceutics-13-01726-f004:**
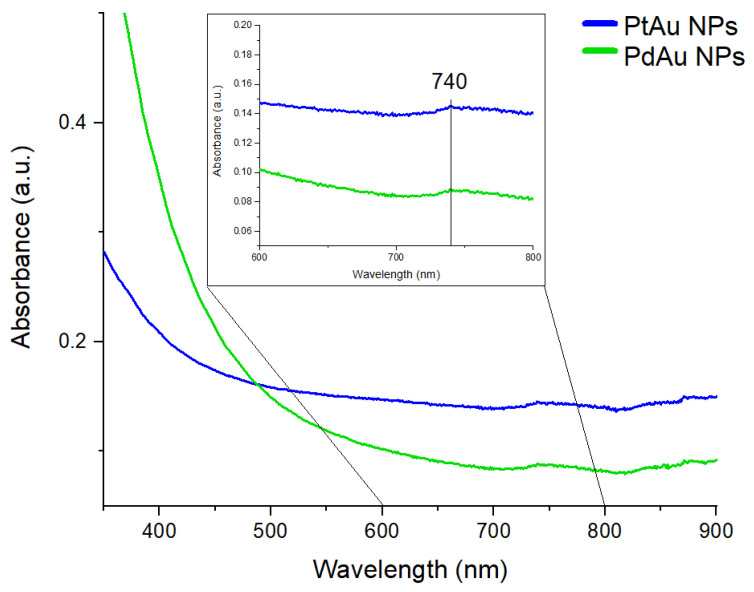
UV-Vis spectra of PtAu NPs (blue curve) and PdAu NPs (green curve).

**Figure 5 pharmaceutics-13-01726-f005:**
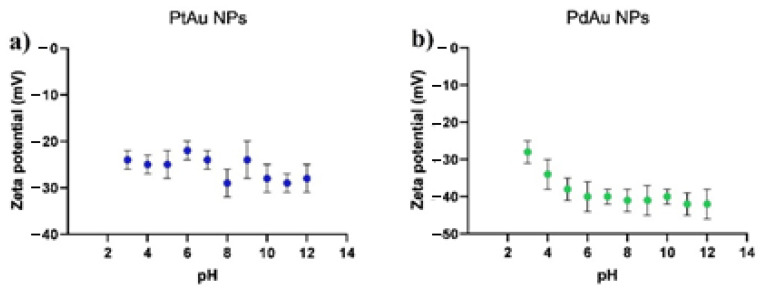
Dependence of zeta potential on the pH for PtAu (**a**) and PdAu NPs (**b**).

**Figure 6 pharmaceutics-13-01726-f006:**
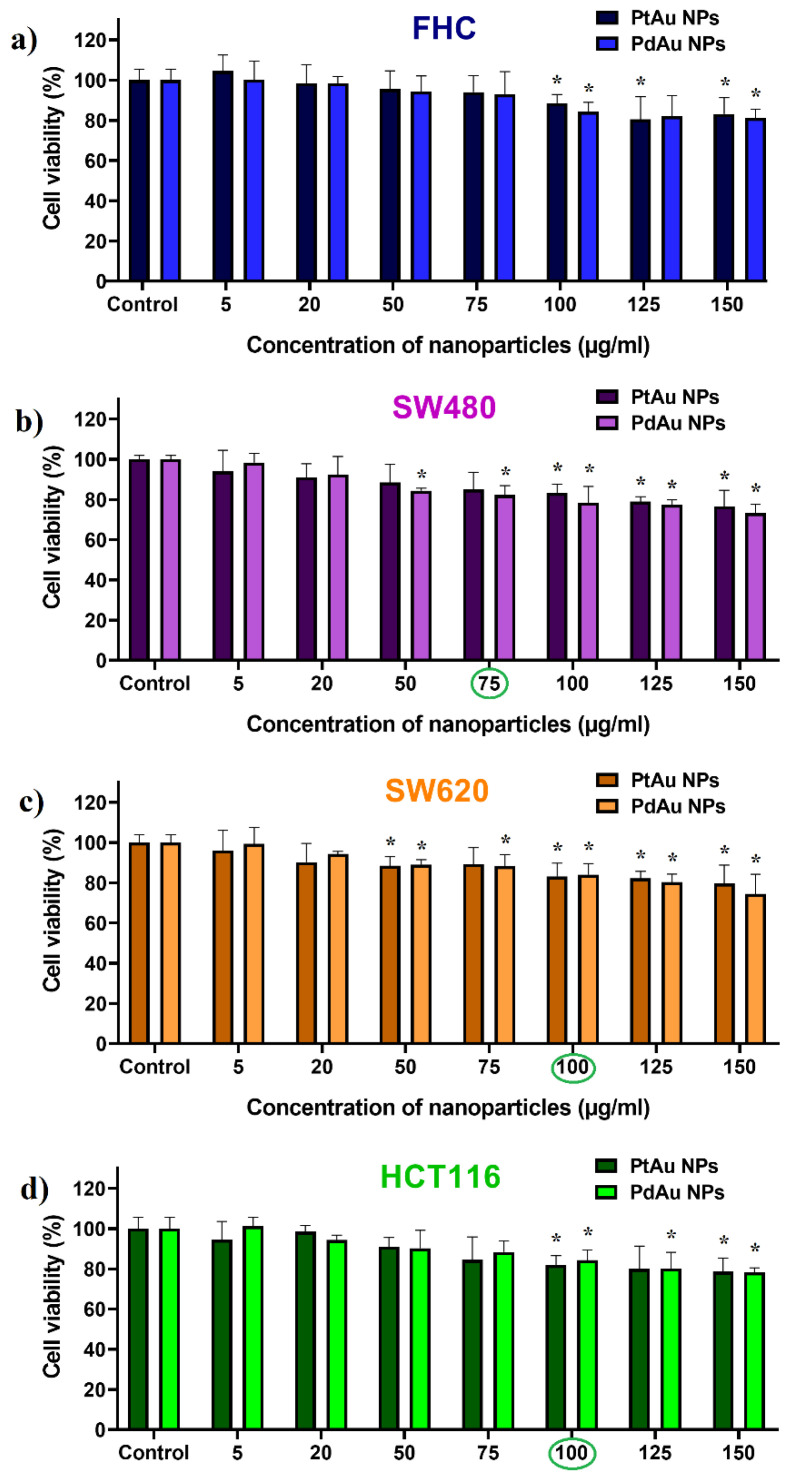
Cytotoxicity of PtAu and PdAu NPs against (**a**) FHC, (**b**) SW480, (**c**) SW620, and (**d**) HCT116 cells after 18 h of incubation. Data were considered significant when * *p* < 0.05 vs. control. The concentration of PtAu and PdAu NPs considered optimal are marked in green.

**Figure 7 pharmaceutics-13-01726-f007:**
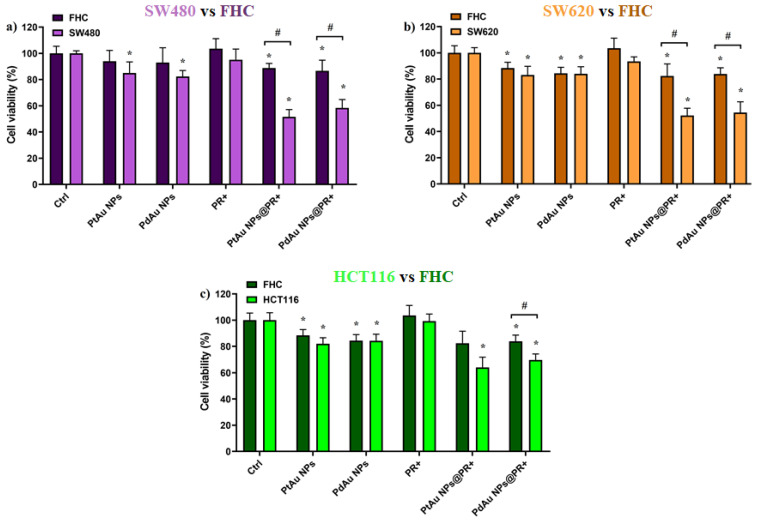
Viability of colon cancer cells (**a**) SW480, (**b**) SW620, and (**c**) HCT116 compared to FHC normal cells after the addition of PtAu/PdAu NPs and subsequent proton beam irradiation with a total dose of 15 Gy combined with PtAu/PdAu NPs. Data were considered significant if * *p* < 0.05 vs. control; # *p* < 0.05—statistically significant differences between the respective cancer lines and the normal FHC line.

**Figure 8 pharmaceutics-13-01726-f008:**
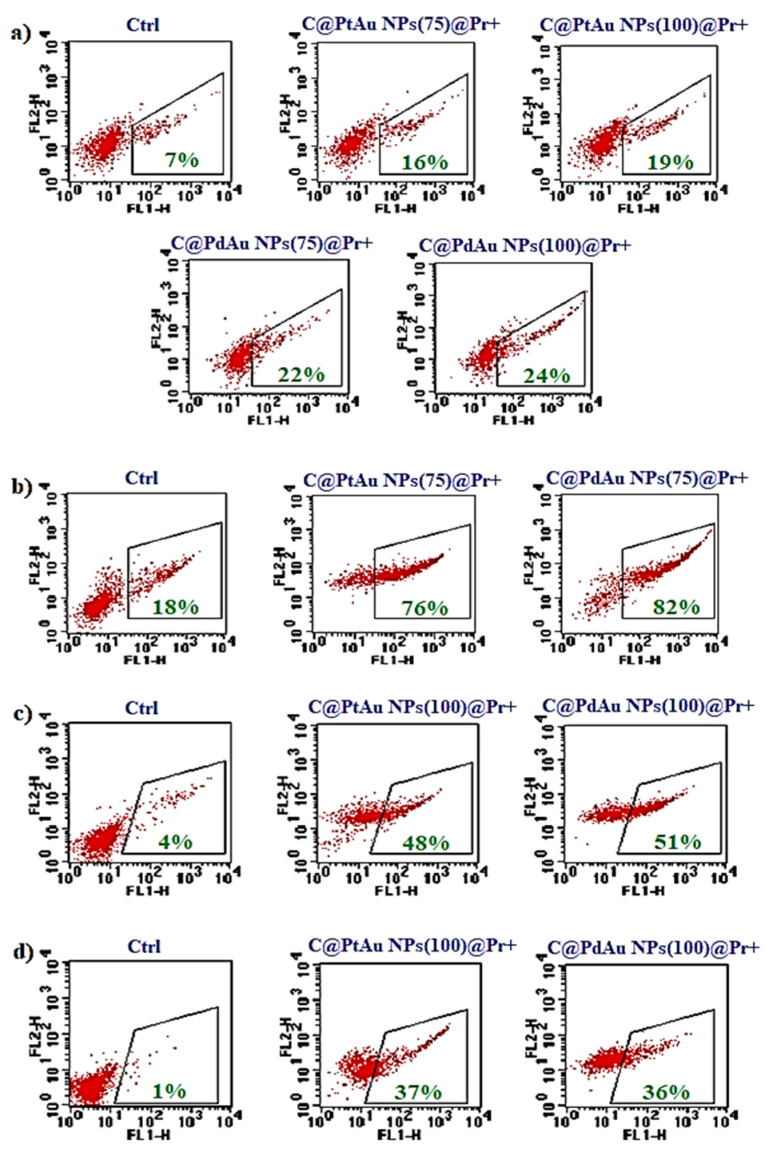
Apoptosis of (**a**) FHC, (**b**) SW480, (**c**) SW620, and (**d**) HCT116 cells determined by Annexin V-binding assay after the treatment with PtAu or PdAu NPs, followed by the proton beam irradiation. Dot-plots show flow cytometry analysis of cells—percent of apoptotic cells (green values) due to the binding of FITC-conjugated Annexin V (FL-1) in comparison to non-treated cells (Ctrl).

**Table 1 pharmaceutics-13-01726-t001:** Description of the investigated samples.

Sample	Name of Sample in the Manuscript
Control samples of cell lines SW480, SW620, HCT116 and FHC (cells without addition PtAu/PdAu NPs and proton irradiation)	Ctrl
Cells cultured with PtAu/PdAu NPs	C@PtAu NPs/C@PdAu NPs
Control samples irradiated by proton beam	C@PR^+^
Cells cultured with PtAu/PdAu NPs and irradiated by proton beam	C@PtAu NPs@PR^+^/C@PdAu NPs@PR^+^

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
