# Peer review of "Gold-Decorated Platinum and Palladium Nanoparticles as Modern Nanocomplexes to Improve the Effectiveness of Simulated Anticancer Proton Therapy"

_pharmaceutics, 2021, doi:10.3390/pharmaceutics13101726_

Round 1
Reviewer 1 Report
In this manuscript, the authors describe gold-decorated platinum and palladium nanoparticles as bimetallic nanocomplexes to improve the effectiveness of simulated anticancer proton therapy.
The application of NPs with a palladium core for proton therapy of cancer is novel approach. From the economic point of view, it was also interesting to show, whether NPs with a palladium core (the synthesis of which is cheaper using gallic acid as reducing agent) will be as effective as NPs with a platinum core. The synthesized nanomaterials were well characterized by scanning transmission electron microscopy, selected area electron diffraction, energy-dispersive X-ray spectroscopy, UV-Vis spectroscopy and zeta potential measurements. The effects of PtAu and PdAu NPs in PT were investigated in vitro using colon cancer cell lines (SW480, SW620 and HCT116), and normal colon epithelium cell line (FHC). The MTS assay showed that the NPs-assisted PT resulted in a better anticancer effect than PT used alone. Flow cytometry analysis using Annexin V showed the cell death as apoptosis. In addition, the data showed that such a treatment was more selective for cancer cells, as normal cell viability was only slightly affected. The application of PtAu and PdAu NPs as potential radiosensitizers in PT of colorectal cancer has shown that such combined approach results in a significant inhibition of cancer cell proliferation and viability. I think the work presented in this manuscript is very useful for readers.
Minor points:
- The titles of 3.2 and 3.3 are the same as 3.1 in the section of Results and discussion, so you should reconsider them.
- P1 Line 14: palladium (Pt NPs) is not (Pd NPs)?
Reviewer 2 Report
The manuscript "Gold-decorated platinum and palladium nanoparticles as modern nanocomplexes to improve the effectiveness of simulated anticancer proton therapy" by Klebowski et al. describes the synthesis of bimetallic nanocomplexes by green chemistry method and their biological characterization with colon cancer cell lines (SW480, SW620 and HCT116), as well as normal colon epithelium cell line (FHC).
The experimental procedures are clearly described, the manuscript as a whole is clear and shows a linear conceptualization of the research. In my opinion it is suitable for publication in this journal.
I have the following comments:
- it would be interesting to investigate the role of ROS in the anticancer effect of the nanocomplexes in colon cancer cells.
- Line 346 Fig 8 caption: specify Pt Au and Pd Au NPs
